# Chaos as an interpretable benchmark for forecasting and data-driven modelling

**William Gilpin**[*]
Department of Physics & Oden Institute, UT Austin
Quantitative Biology Initiative, Harvard University
wgilpin@fas.harvard.edu

## Abstract

The striking fractal geometry of strange attractors underscores the generative nature of chaos: like probability distributions, chaotic systems can be repeatedly measured to produce arbitrarily-detailed information about the underlying attractor. Chaotic systems thus pose a unique challenge to modern statistical learning techniques, while retaining quantifiable mathematical properties that make them controllable and interpretable as benchmarks. Here, we present a growing database currently comprising 131 known chaotic dynamical systems spanning fields such as astrophysics, climatology, and biochemistry. Each system is paired with precomputed multivariate and univariate time series. Our dataset has comparable scale to existing static time series databases; however, our systems can be re-integrated to produce additional datasets of arbitrary length and granularity. Our dataset is annotated with known mathematical properties of each system, and we perform feature analysis to broadly categorize the diverse dynamics present across the collection. Chaotic systems inherently challenge forecasting models, and across extensive benchmarks we correlate forecasting performance with the degree of chaos present. We also exploit the unique generative properties of our dataset in several proof-of-concept experiments: surrogate transfer learning to improve time series classification, importance sampling to accelerate model training, and benchmarking symbolic regression algorithms.

## 1 Introduction

Two trajectories emanating from distinct locations on a strange attractor will never recur nor intersect, a basic mathematical property that underlies the complex geometry of chaos. As a result, measurements drawn from a chaotic system are deterministic yet non-repeating, even at finite resolution [1, 2]. Thus, while representations of chaotic systems are finite (e.g. differential equations or discrete maps), they can indefinitely generate new data, allowing the fractal structure of the attractor to be resolved in ever-increasing detail [3]. This interplay between the boundedness of the attractor and non-recurrence of the dynamics is responsible for the complexity of diverse systems, ranging from the intricate gyrations of orbiting stars to the irregular spiking of neuronal ensembles [4, 5].

Chaotic systems thus represent a unique testbed for modern statistical learning techniques. Their unpredictability challenges traditional forecasting methods, while their fractal geometry precludes concise representations [6]. While modeling and forecasting chaos remains a fundamental problem in its own right [7, 8], many prior works on general time series analysis and data-driven model inference have used specific chaotic systems (such as the Lorenz "butterfly" attractor) as toy problems in order to demonstrate method performance in a controlled setting [9–20]. In this context, there are several advantages to chaotic systems as benchmarks for time series analysis and data-driven modelling:

---

[*]Dataset and benchmark available at: https://github.com/williamgilpin/dysts

35th Conference on Neural Information Processing Systems (NeurIPS 2021) Track on Datasets and Benchmarks.

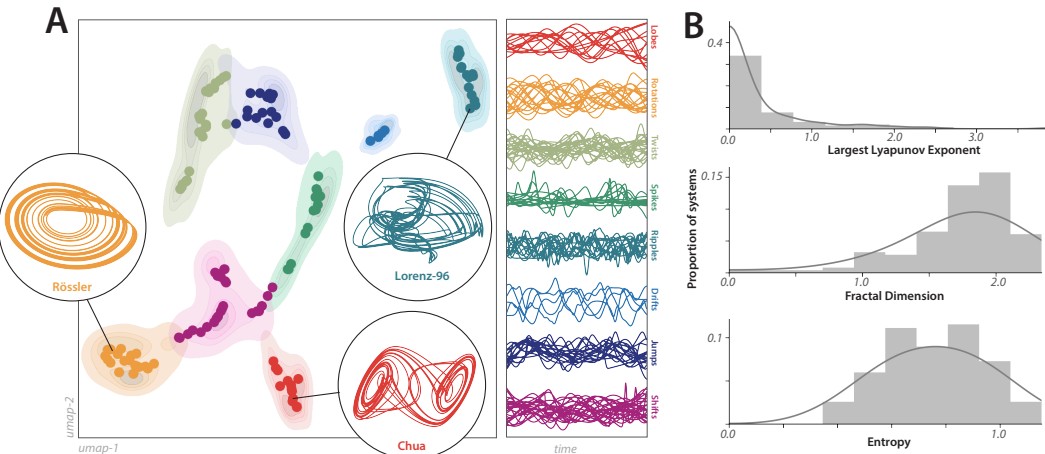

Figure 1: **Properties of the chaotic dynamical systems dataset.** (A) Embeddings of 131 chaotic dynamical systems. Points correspond to average embeddings of individual systems, and shading shows ranges over many random initial conditions. Colors correspond to an unsupervised clustering, and example dynamics for each cluster are shown. (B) Distributions of key mathematical properties across the dataset.

(1) Chaotic systems have provably complex dynamics, which arise due to underlying mathematical structure, rather than abstruse representations of otherwise simple latent dynamics.
(2) Existing time series databases contain datasets chosen primarily for availability or applicability, rather than for having innate properties (e.g. complexity, quasiperiodicity, dimensionality) that span the range of possible behaviors time series may exhibit.
(3) Chaotic systems have accessible generating processes, making it possible to obtain new data and representations, and for the benchmark to be related to mechanistic details of the underlying system. These properties suggest that chaotic systems can aid in interpreting the properties of complex models [6, 21]. However, chaotic systems as benchmarks lack standardization, and prior works' emphasis on single systems like the Lorenz attractor may undermine generalizability. Moreover, focusing on isolated systems neglects the diversity of dynamics in known chaotic systems, thereby preventing systematic quantification and interpretation of algorithm performance relative to the mathematical properties of different systems.

Here, we present a growing database of low-dimensional chaotic systems drawn from published work in diverse domains such as meteorology, neuroscience, hydrodynamics, and astrophysics. Each system is represented by several multivariate time series drawn from the dynamics, annotations of known mathematical properties, and an explicit analytical form that can be re-integrated to generate new time series of arbitrary length, stochasticity, and granularity. We provide extensive forecasting benchmarks across our systems, allowing us to interpret the empirical performance of different forecasting techniques in the context of mathematical properties such as system chaoticity. Our dataset improves the interpretability of time series algorithms by allowing methods to be compared across time series with different intrinsic properties and underlying generating processes—thereby complementing existing interpretability methods that identify salient feature sets or time windows within single time series [22, 23]. We also consider applications to data-driven modelling in the form of symbolic regression and neural ordinary differential equations tasks, and we show the surprising result that the accuracy of a symbolic regression-derived formula can correlate with mathematical properties of the dynamics produced by the formula. Finally, we demonstrate unique applications enabled by the ability to re-integrate our dataset: we pre-train a timescale-matched feature extractor for an existing time series classification benchmark, and we accelerate training of a forecast model by importance sampling sparse regions on the dynamical attractor.

## 2    Description of Datasets

**Scope.**    The diverse dynamical systems in our dataset span astrophysics, neuroscience, ecology, climatology, hydrodynamics, and many other domains. The supplementary material contains a glossary defining key terms from dynamical systems theory relevant to our dataset. Each entry in

our dataset represents a single dynamical system, such as the Lorenz attractor, that takes an initial condition as input and outputs a trajectory representing the input point's location as time evolves. Systems are chosen based on prior appearance as named systems in published works. In order to provide a consistent test for time series models, we define chaos in the mathematical sense: two copies of a system prepared in infinitesimally different initial states will exponentially diverge over time. We also focus particularly on chaotic systems that produce low-dimensional strange attractors, which are fractal structures that display bounded, stationary, and ergodic dynamics with quantifiable mathematical properties. As a result, we exclude transient chaos and chaotic *repellers* (chaotic regions that trajectories eventually escape) [24–26], as well as most nonchaotic strange attractors save for one paradigmatic example: a quasiperiodic two-dimensional torus [27].

**Scale and structure.**    Our extensible collection currently comprises 131 previously-published and named chaotic dynamical systems. Each record includes a compilable implementation of the system, a citation reference, default initial conditions on the attractor, precomputed train and test trajectories from different initial conditions at both coarse and fine granularities, and an optimal integration timestep and dominant timescale (used for aligning timescales across systems). For each of the 131 systems, we include 16 precomputed trajectories corresponding to all combinations of the following variations per system: coarse and fine sampling granularity, train and test splits emanating from different initial conditions, multivariate and univariate views, and trajectories with and without Brownian noise influencing the dynamics. Because certain data-driven modelling methods, such as our symbolic regression task below, require gradient information, we also include with each system precomputed train and test regression datasets corresponding to trajectories and time derivatives along them.

Figure S1 shows the attractors for all systems, and Table S1 includes brief summaries of their origin and applications. While there are an infinite number of possible chaotic dynamical systems, our work represents, to our knowledge, the first effort to survey and reproduce previously-published chaotic systems. For this reason, while our dataset is readily extensible to new systems, the primary bottleneck as we expand our database is the need to manually reproduce claimed chaotic dynamics, and to identify appropriate parameter values and initial conditions based on published reports. Broadly, our work can be considered a systematization of previous studies that benchmark methods on single chaotic systems such as the Lorenz attractor [9–20].

**Annotations.**    For each system, we calculate and include precise estimates of several standard mathematical characteristics of chaotic systems. More detailed definitions are included in the appendix.

*The largest Lyapunov exponent* measures the degree to which nearby trajectories diverge, a common measure of the degree of chaos present in a system.

*The Lyapunov exponent spectrum* determines the tendency of trajectories to locally converge or diverge across the attractor. All continuous-time chaotic systems have at least one positive exponent, exactly one zero exponent (due to time translation), and, for dissipative systems (i.e., those converging to an attractor), at least one negative exponent [28].

*The correlation dimension* measures an attractor's effective fractal dimension, which informally indicates the intricacy of its geometric structure [4, 29]. Integer fractal dimensions indicate familiar geometric forms: a line has dimension one, a plane has two, and a filled solid three. Non-integer values correspond to fractals that fill space in a manner intermediate to the two nearest integers.

*The multiscale entropy* represents the degree to which complex dynamics persist across timescales [30]. Chaotic systems have continuous power spectra, and thus high multiscale entropy.

We also include two quantities derived from the Lyapunov spectrum: *the Pesin entropy bound*, and *the Kaplan-Yorke fractal dimension*, an alternative estimator of attractor dimension based on trajectory dispersion. Each system is also annotated with various qualitative details, such as whether the system is Hamiltonian or dissipative (i.e., whether there exists conserved invariants like total energy, or whether the dynamics relax to an attractor), non-autonomous (whether the dynamical equations explicitly depend on time), bounded (all variables remain finite as time passes), and whether the dynamics are given by a delay differential equation. In addition to the 131 differential equations described here, our collection also includes several common discrete time maps; however, we exclude these from our study due to their unique properties.

**Methods.** Our dataset includes utilities for re-sampling and re-integrating each system with or without stochasticity, loading pre-computed multivariate or univariate trajectories, computing statistical properties and performing surrogate significance testing, and running benchmarks. One shortcoming of previous studies using chaotic systems as benchmarks—as well as more generally with static time series databases—is inconsistent timescales and granularities (sampling rates). We alleviate this problem by using phase surrogate significance testing to select optimal integration timesteps and sampling rates for all systems in our dataset, thus ensuring that dynamics are aligned across systems with respect to dominant and minimum significant timescales [21]. We further ensure consistency across systems using several standard methods, such as testing ergodicity to find consistent initial conditions, and integrating with continuous re-orthonormalization when computing various mathematical quantities such as Lyapunov exponents (see supplementary material).

**Properties and Characterization.** In order to characterize the properties of our collection, we use an off-the-shelf time series featurizer that computes a corpus of 787 common time series features (e.g. absolute change, peak count, wavelet transform coefficients, etc) for each system in our dataset [31]. In addition to providing general statistical descriptors for our systems, embedding and clustering the systems based on these features illustrates the diverse dynamics present across our dataset (Figure 1). We find that the dynamical systems naturally separate into groups displaying different types of chaotic dynamics, such as smooth scroll-like trajectories versus spiking. Additionally, we observe that our chaotic systems trace a filamentary manifold in embedding space, a property consistent with the established rarity of chaotic attractors within the space of possible dynamical systems: persistent chaos often occurs in an intermediate regime between bifurcations producing simpler dynamics, such as limit cycles or quiescence at fixed points [5, 26].

## 2.1 Prior Work.

**Data-driven modelling and control.** Many techniques at the intersection of machine learning and dynamical systems theory have been evaluated on specific well-known chaotic attractors, such as the Lorenz, Rössler, double pendulum, and Chua systems [9–20]. These and several other chaotic systems used in previous machine learning studies are all included within our dataset [32, 33]. General databases of analytical mathematical models include the BioModels database of systems biology models, which currently contains 1017 curated entries, with an additional 1271 unreviewed user submissions [34]. Among these models, a subset corresponding to 491 differential equations appear within the ODEBase database [35]. For the specific task of symbolic regression, the inference of analytical equations from data, existing benchmarks include the Nguyen dataset of 12 complex mathematical expressions [36], and corpora of equations from two physics textbooks [37–39], and a recently-released suite of 252 regression problems from Penn Machine Learning Benchmark [40].

**Forecasting and classification of time series.** The UCR-UEA time series classification benchmark includes 128 univariate and 30 multivariate time series with ~$10^1$–$10^3$ timepoints [41–44]. Several of these entries overlap with the UCI Machine Learning Repository, which contains 121 time series (91 multivariate) of lengths ~$10^1$–$10^6$ [45]. The M-series of time series forecasting competitions have most recently featured $10^6$ univariate time series of length ~$10^1$–$10^5$ [46]. The recently-introduced Monash forecasting archive comprises 26 domain areas, each of which includes ~$10^1$–$10^6$ distinct time series with lengths in the range ~$10^2$–$10^6$ timepoints [47]. A recent long-sequence forecasting model uses the `ETT-small` dataset of electricity consumption in two regions of China (70,080 datapoints at one-minute increments) [48], as well as NOAA local climatological data (~$10^6$ hourly recordings from ~$10^3$ locations) [49]. The PhysioNet database contains several hundred physiological recordings such as EEG, ECG, and blood pressure, at a wide variety of resolutions and lengths [50].

A point of differentiation between our work and existing datasets is our focus on reproducible chaotic dynamics, which sufficiently narrows the space of potential systems that we can manually curate and re-implement reported dynamics, and calculate key mathematical properties relevant to forecasting and physics-based model inference. These mathematical properties can be used to interpret the properties of black box models by examining their correlation with model performance across systems. Our dataset's curation also ensures a high degree of standardization across systems, such as consistent integration and sampling timescales, as well as ergodicity and stationarity. Additionally, the precomputed multivariate time series in our dataset approximately match the length and size of existing time series databases. We emphasize that, unlike existing time series databases, our

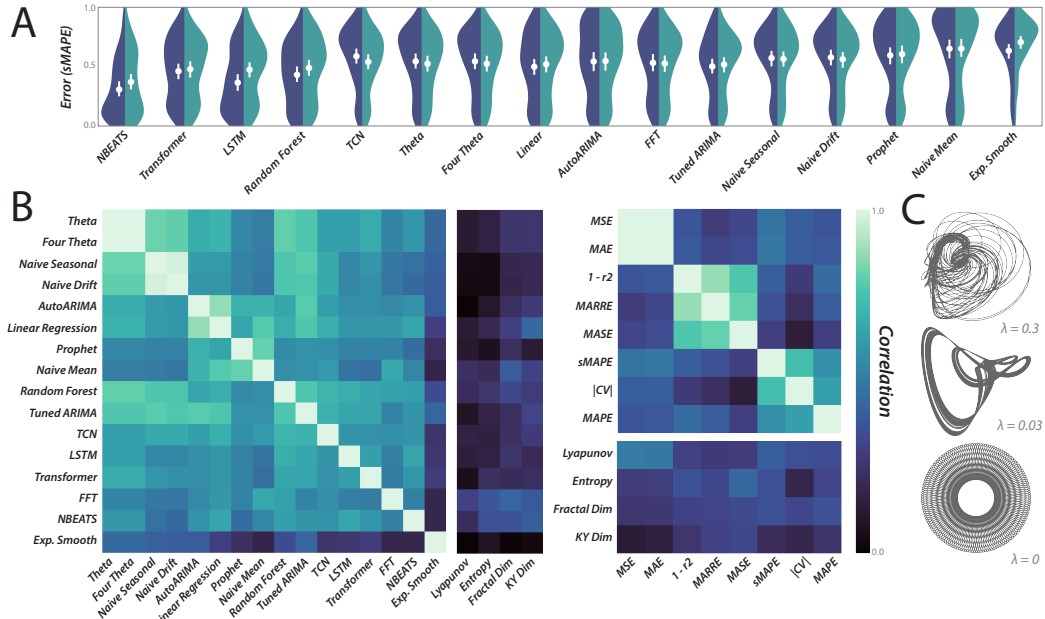

Figure 2: **Forecasting benchmarks for all chaotic dynamical systems.** (A) Distribution of forecast errors for all dynamical systems and for all forecasting models, sorted by increasing median error. Dark and light hues correspond to coarse and fine time series granularities. (B) Spearman correlation among forecasting models, among different forecast evaluation metrics, and between forecasting metrics and underlying mathematical properties, computed across all dynamical systems at fine granularity. Columns are ordered by descending maximum cross-correlation in order to group similar models and metrics. (C) The systems with the highest, median, and lowest forecasting error across all models, annotated by largest Lyapunov exponent.

dataset's size is flexible due to the ability to re-integrate each system at arbitrary length, sample at any granularity, integrate from new initial conditions, change the amount of stochastic forcing, or even perturb parameters in the underlying differential equation in order to modify or control each system's dynamics.

## 3   Experiments

**Task 1: Forecasting**

Chaotic systems are inherently unpredictable, and extensive work by the physics community has sought to quantify chaos, and to relate its properties to general features of the underlying governing equations [5, 6]. Traditionally, the predictability of a chaotic system is thought to be determined by the largest Lyapunov exponent, which measures the rate at which trajectories emanating from two infinitesimally-spaced points will exponentially separate over time [28].

We evaluate this claim on our dataset by benchmarking 16 forecasting models spanning a wide variety of techniques: deep learning methods (NBEATS, Transformer, LSTM, and Temporal Convolutional Network), statistical methods (Prophet, Exponential Smoothing, Theta, 4Theta), common machine learning techniques (Random Forest), classical methods (ARIMA, AutoARIMA, Fourier transform regression), and standard naive baselines (naive mean, naive seasonal, naive drift) [47, 51–53]. Our train and test datasets correspond to differential initial conditions, and we perform separate hyperparameter tuning for each chaotic system and granularity [53, 54]. While the forecasting models are heterogenous, for each we tune whichever hyperparameter most closely corresponds to a timescale—for example, the lag order for autoregressive models, or the input chunk size for the neural network models. Because all systems are aligned to the same average period, the range of values over which timescales are tuned is scaled by the granularity. Hyperparameters are tuned using

held-out future values, and scores are computed on an unseen test trajectory emanating from different initial conditions.

Our results are shown in Figure 2 for all dynamical systems at coarse and fine sampling granularity. We include corresponding results for systems with noise in the supplementary material. We find the the deep learning models perform particularly well, with the Transformer and NBEATS models achieving the lowest median scores, while also appearing within the three best-performing models for nearly all systems. On many datasets, the temporal convolutional network and traditional LSTM models also achieve competitive performance. Notably, the random forest also exhibits strong performance despite the continuous nature of our datasets, and with substantially lower training cost. The relative ranking of the different forecasting models remains stable both as granularity is varied over two orders of magnitude, and as noise is increased to a level dominating the signal (see supplementary experiments). In the latter case, we observe that the performance of different models converges as their overall performance decreases. Overall, NBEATS strongly outperforms the other forecasting techniques across varied systems and granularities, and its performance persists even in the presence of noise. We speculate that NBEAT's advantage arises from its implicit decomposition of time series into a hierarchy of basis functions [51], an approach that mirrors classical techniques for representing continuous-time chaotic systems [55].

Our results seemingly contrast with studies showing that statistical models outperform neural networks on forecasting tasks [46, 47]. However, our forecasting task focuses on long time series and prediction horizons, two areas where neural networks have previously performed well [48]. Additionally, we hypothesize that the strong performance of deep learning models on our dataset is a consequence of the smoothness of chaotic systems, which have mathematical regularity and stationarity compared to time series generated from industrial or environmental measurements. In contrast, models like Prophet are often applied to calendar data with seasonality and irregularities like holidays [56]—neither of which have a direct analogue in chaotic systems, which contain a continuous spectrum of frequencies [57]. Consistent with this intuition, we observe that among the systems in our dataset, the Prophet model performs well on the torus, a quasiperiodic system with largest Lyapunov exponent equal to zero.

Several recent works have considered the appropriate metric for determining forecast accuracy [46, 47, 58, 59]. For all forecasting models and dynamical systems we compute eight error metrics: the mean squared error (MSE), mean absolute scaled error (MASE), mean absolute error (MAE), mean absolute ranged relative error (MARRE), the magnitude of the coefficient of variation ($|CV|$), one minus the coefficient of determination ($1 - r^2$), and the symmetric and regular mean absolute percent errors (MAPE and sMAPE). We find that all of these potential metrics are positively correlated across our dataset, and that they can be grouped into families of strongly-related metrics (Figure 2B). We also observe that the relative ranking of different forecasting models is independent of the choice of metric. Hereafter, we report sMAPE errors when comparing models, but we include all other metrics within the benchmark.

We next evaluate the common claim that the empirical predictability of a system depends on the mathematical degree of chaos present [7]. For each system, we correlate the forecast error of the best-performing model with the various mathematical properties of each system (Figure 2B). Across all systems, we find a high degree of correlation between the largest Lyapunov exponent and the forecast error, while other measures such as the attractor fractal dimension and entropy correlate less strongly. While this observation matches conventional wisdom, it represents (to our knowledge) the first large-scale test of the empirical relevance of Lyapunov exponents. We consider this observation particularly noteworthy because our forecasting task spans several periods, yet the Lyapunov exponent measures only the local dispersion between infinitesimally-separated points.

Our results introduce several considerations for the development of time series models. The strong performance we observe for neural network models implies that the flexibility of large models proves beneficial for time series without obvious trends or seasonality. The consistent accuracy we observe for NBEATS, even in the presence of noise, suggests that hierarchical decomposition can improve modelling of systems with multiple timescales. Most of our best-performing methods implicitly lift the dimensionality of the input time series, implying that higher-dimensional representations create more predictable dynamics—a finding consistent with recent studies showing that certain machine learning techniques implicitly learn Koopman operators, linear propagators that act on lifted representations of nonlinear systems [10, 57, 60–62]. That higher dimensional representations can

Table 1: (Upper) Forecast accuracy for LSTMs trained on full time series, random subsets, and subsets sampled proportionately to their epochwise error (medians ± standard errors across all dynamical systems). (Lower) Accuracy scores on the UCR database for classifiers trained on features extracted from bare time series, and from autoencoders pretrained on the full chaotic systems collection at random and task-matched timescales (medians ± standard errors across UCR tasks).

| Importance Sampling Forecasting Error (sMAPE) | | | |
| --- | --- | --- | --- |
| | Full Epochs | Random Subset | Importance Weighted |
| sMAPE | $1.00 \pm 0.05$ | $0.99 \pm 0.05$ | $\mathbf{0.90 \pm 0.05}$ |
| Runtime (sec) | $190.1 \pm 0.3$ | $\mathbf{77.9 \pm 0.3}$ | $94.6 \pm 0.2$ |

| Transfer Learning Classification Accuracy | | | |
| --- | --- | --- | --- |
| | No Transfer Learning | Random Timescales | Matched Surrogates |
| Accuracy | $0.80 \pm 0.02$ | $0.82 \pm 0.01$ | $\mathbf{0.84 \pm 0.01}$ |

linearize dynamics mirrors classical motivation for kernel methods in machine learning [63]; we thus hypothesize that classical time series representations like time-lagged embeddings can be improved through nonlinearities, either in the form of custom functions learned by neural networks, or by inductive biases in the form of fixed periodic or wavelet-like kernels.

**Task 2: Accelerating model training with importance sampling.**

When training a forecasting model iteratively, each training batch usually samples input timepoints with equal probability. However, chaotic attractors generally possess non-uniform measure due to their fractal structure [64]. We thus hypothesize that importance sampling can accelerate training of a forecast model, by encouraging the network to oversample sparser regions of the underlying attractor [65]. We thus modify the training procedure for a forecast model by applying a simple form of importance sampling, based on the epoch-wise training losses of individual samples—an approach related to zeroth-order adaptive methods appearing in other areas [66–69]. Our procedure consists of the following: (1) We halt training every few epochs and compute historical forecasts (backtests) on the training trajectory. (2) We randomly sample timepoints proportionately to their error in the historical forecast, and then generate a set of initial conditions corresponding to random perturbations away from each sampled attractor point. (3) We simulate the full dynamical system for $\tau$ timesteps for each of these initial conditions, and we use these new trajectories as the training set for the next $b$ epochs. We repeat this procedure for $\nu$ meta-epochs. For the original training procedure, the training time scales as $\sim B$, the number of training epochs. In our modified procedure, the training time has dominant term $\sim \nu\, b$, plus an additional term proportional to $\tau$ (integration can be parallelized across initial conditions), plus a small constant cost for sampling. We thus set $\nu\, b < B$, and record run times to verify that total cost has decreased.

Table 1 shows the results of our experiments for an LSTM model across all chaotic attractors. Importance sampling achieves a significantly smaller forecast error than a baseline using the full training set in each epoch, as well as a control in which the exact importance sampling procedure was repeated without weighting random samples by error (two sided paired t-test, $p < 10^{-6}$ for both tests). Notably, importance sampling requires substantially lower computation due to the reduced number of training epochs incurred. Our approach exploits that our database comprises strange *attractors*, because initial conditions derived from random perturbations off an attractor will produce trajectories that return to the attractor.

**Task 3: Transfer learning and data augmentation.**

We next explore how our dataset can assist general time series analysis, regardless of the relevance of chaos to the problem. We study an existing time series classification benchmark, and we use our dataset to generate timescale-matched surrogate data for transfer learning.

Our classification procedure broadly consists of training an autoencoder on trajectories from our database, and then using the trained encoder as a general feature extractor for time series classification. However, unlike existing transfer learning approaches for time series [70], we train the autoencoder

on a new dataset for each classification problem: we re-integrate our entire dataset to match the dominant timescales in the classification problem's training data.

Our approach thus comprises several steps: (1) Across all data in the train partition, the dominant significant Fourier frequency is determined using random phase surrogates [21]. (2) Trajectories are re-integrated for every dynamical system in our database, such that the sampling rate of the dynamics is equal to that of the training dataset. The surrogate ensemble thus corresponds to a custom set of trajectories with timescales matched to the training data of the classification problem. (3) We train an autoencoder on this ensemble. Our encoder is a one layer causal dilated encoder with skip connections, an architecture recently shown to provide strong time series classification performance [71]. (3) We apply the encoder to the training data of the classification problem. (4) We apply a standard linear time series classifier, following recent works [43, 72]. We featurize the time series using a library of standard featurizers [31], and then perform classification using ridge regression [72]. Overall, our classification approach bears conceptual similarity to other generative data augmentation techniques: we extract parameters (the dominant timescales) from the training data, and then use these parameters to construct a custom surrogate ensemble with matching timescales. In many image augmentation approaches, a prior distribution is learned from the training data (e.g. via a GAN), and the then sampled to create surrogate examples [73–76].

As baselines for our approach, we train a classifier on the bare original time series, as well as a "random timescale" collection in which the time series in the surrogate ensemble have random dominant frequencies, unrelated to the timescales in the training data. The latter ablation serves to isolate the role of timescale matching, which is uniquely enabled by the ability to re-integrate our dataset at arbitrary granularity. This control experiment is necessary in light of recent work showing that transfer learning on a large collection of time series can yield informative features [70].

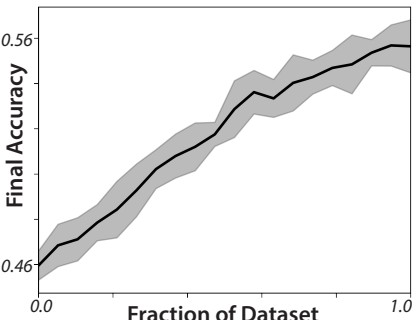

Figure 3: Classification accuracy on the UCR dataset `EOGHorizontalSignal`, across models pretrained on increasing fractions of the database. Standard errors are from bootstrapped replicates, where the dynamical systems are sampled with replacement.

We benchmark classification using the UCR time series classification benchmark, which contains 128 real-world classification problems spanning diverse areas like medicine, agriculture, and robotics [41]. Because we are using convolutional models, we restrict our analysis to the 91 datasets with at least 100 contiguous timepoints (these include the 85 "bakeoff" datasets benchmarked in previous studies) [42]. We compute separate benchmarks (surrogate ensembles, features, and scores) for each dataset in the archive.

Our results are shown in Table 1. Across the UCR archive we observe statistically significant average classification accuracy increases of $4\% \pm 1\%$ compared to the raw dataset ($p < 10^{-4}$, paired two-sided t-test), and $2\% \pm 1\%$ compared to the ablation with random surrogate timescales ($p < 10^{-4}$). While these modest improvements do not comprise state-of-the-art results on the UCR database [42], they demonstrate that features learned from chaotic systems in an unsupervised setting can be used to extract meaningful general features for further analysis. On certain datasets, our results approach other recent unsupervised approaches in which a simple linear classifier is trained on top of a complex unsupervised feature extractor [43, 70, 71]. Recent results have even shown that a very large number of *random* convolutional features can provide informative representations of time series for downstream supervised learning tasks [43]; we therefore speculate that pretraining with chaotic systems may allow more efficient selection of informative convolutional kernels. Moreover, the improvement of transfer learning over the random timescale model demonstrates the advantage of re-integration. In order to verify that the diversity of dynamical systems present within our dataset contribute to the quality of the learned features, we repeat the classification task on a single UCR dataset, corresponding to clinical eye tracking data. We train encoders on gradually increasing numbers of dynamical systems, in order to see how the final accuracy changes as the number of systems available for pretraining increases (Figure 3). We observe monotonic scaling, indicating that our dataset's size and diversity contribute to feature quality.

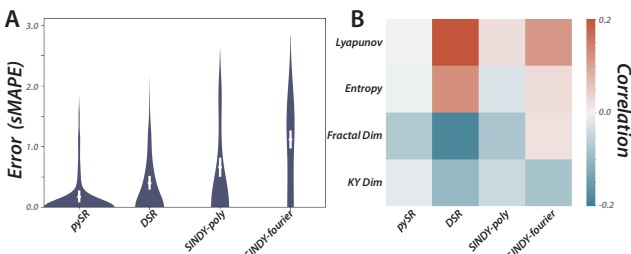

Figure 4: **Symbolic regression benchmarks.** (A) Error distributions on test datasets across all systems, (B) Spearman correlation between errors and mathematical properties of the underlying systems.

**Task 4: Data-driven model inference and symbolic regression**

We next look beyond traditional time series analysis, and apply our database to a data-driven modelling task. A growing body of work uses machine learning methods to infer dynamical systems directly from data [77–80]. Examples include constructing effective propagators for the dynamics [57, 61, 62, 81–83], obtaining neural representations of the dynamical equations [84–87], and inferring analytical governing equations via symbolic regression [37, 40, 60, 88–93]. Beyond improving forecasts, these data-driven representations can discover mechanistic insights, such as symmetries or separated timescales, that might not otherwise be apparent in a time series.

We thus use our dataset for data-driven modelling in the form of a symbolic regression task. We focus on symbolic regression because of the recent emergence of widely-used benchmark models and performance desiderata for these methods [40]. However, we emphasize that our database can be used for other emerging focus areas in data-driven modelling, such as inference of empirical propagators or neural ordinary differential equations [57, 82, 94], and we include a baseline neural ordinary differential equation task in the supplementary material. For each dynamical system in our collection, we generate trajectories with sufficiently coarse granularity to sample all regions of the attractor. At each timepoint, we compute the value of the right hand side of the corresponding dynamical equation, and we treat the value of this time derivative as the regression target. We use this dataset to compare several recent symbolic regression approaches: (1) DSR: a recurrent neural network trained with a risk-seeking policy gradient, which produces state-of-the-art results on a variety of challenging symbolic regression tasks [88]. (2) PySR: an open-source package inspired by the popular closed-source software Eureqa, which uses genetic programming and simulated annealing [90, 92, 95]. (3,4) PySINDY: a Python implementation of the widely-used SINDY algorithm, which uses sparse regression to decompose data into linear combinations of functions [89, 96]. For PySINDY we train separate models for purely polynomial (SINDY-poly) and trigonometric (SINDY-fourier) bases. For DSR and pySR we use a standard library of binary and unary expressions, $\{+, -, \times, \div\}$, $\{\sin, \cos, \exp, \log, \tanh\}$ [88]. After fitting a formula using each method, we evaluate it on an unseen test trajectory arising from different initial conditions, and we report the the sMAPE error between the formula's prediction and the true value along the trajectory.

Our results illustrate several features of our dataset, while also illustrating properties of the different symbolic regression algorithms. All algorithms show strong performance across the chaotic systems dataset (Figure 4). The two lowest-error models, pySR and DSR, exhibit nearly-equivalent performance when accounting for error ranges, and both achieve errors near zero on many systems. We attribute this strong performance to the relatively simple algebraic construction of most published systems: named chaotic systems will inevitably favor concise, demonstrative equations over complex expressions. In fact, several systems in our dataset belong to the Sprott attractor family, which represent the algebraically simplest chaotic systems [97]. In this sense, our dataset likely has similar complexity to the Feynman equations benchmark [37].

We highlight that PySINDY with a purely polynomial basis performs very well on our dataset, especially as a linear approach that requires a median training time of only $0.01 \pm 0.01$ s per system on a single CPU core. In comparison, pySR had a median time of $1400 \pm 60$ s per system on one core, while DSR on one GPU required $4300 \pm 200$s per system—consistent with the results of recent symbolic regression benchmark suite [40]. However, parallelization reduces the runtime of all methods.

We emphasize that the relative performance of a given symbolic regression algorithm depends on diverse factors, such as equation complexity, the library of available unary and binary operators, the amount and dynamic range of available input data, the amount of compute available for refinement, and the degree of nonlinearity of the underlying system. More generally, symbolic regression algorithms exhibit a bias-variance tradeoff manifesting as Pareto front bridging accuracy and parsimony: large models with many terms will appear more accurate, but at the expense of brevity and potentially robustness and interpretability [90]. More challenging benchmarks would include nested expressions and uncommon transcendental functions; these systems may be a more appropriate setting for benchmarking state-of-the-art techniques like DSR. Additionally, we do not include measurement noise in our experiments, a scenario in which DSR performs strongly compared to other methods [40, 88].

Interestingly, DSR exhibits the strongest dependence on the mathematical properties of the underlying dynamics: more chaotic systems consistently yield higher errors (Figure 4B). We consider this result surprising, because *a priori* we would expect the performance of a given symbolic regression algorithm to depend purely on the syntactic complexity of the target formula, rather than the dynamics that it produces. Because DSR uses a large model to navigate a space of smaller models, we hypothesize that more chaotic systems present a broader set of possible "partial formulae" that match specific subregimes of the attractor—an effect exploited in several recent decomposition techniques for chaotic systems [9, 11]. The diversity of these local approximants would result in a more complex global search space.

# 4   Discussion

We have introduced an extensible collection of known chaotic dynamical systems. In addition to representing a customizable benchmark for time series analysis and data-driven modelling, we have provided examples of additional applications, such as transfer learning for general time series analysis tasks, that are enabled by the generative nature of our dataset. We note that there are several other potential applications that we have not explored here: testing feedback-based control algorithms (which require perturbing the parameters of a given dynamical system, and then re-integrating), and inferring numerical propagators (such as Koopman operators)[57, 61, 62, 81–83, 98, 99]. In the appendix, we include preliminary benchmarks for a neural ordinary differential equations task [84–87]; due to the direct connections between our work and this area, we hope to further explore these methods in future studies. Our work can be seen as systematizing the common practice of testing new methods on single chaotic systems, particularly the Lorenz attractor [9–20].

More broadly, our collection seeks to improve the interpretability of data-driven modelling from time series. For example, our forecasting benchmark experiments show that the Lyapunov exponent, a popular measure of chaoticity, correlates with the empirical predictability of a system under a variety of models—a finding that matches intuition, but which has not (to our knowledge) previously been tested extensively. Likewise, in our symbolic regression benchmark we find that more chaotic systems are harder to model, an effect we attribute to the diverse local approximants available for complex dynamical systems. These examples demonstrate how the control and mathematical context provided by differential equations can yield mechanistic insight beyond traditional time series.

Limitations of our approach include our inclusion only of known chaotic systems that have previously appeared in published works. This limits the rate at which our collection may expand, since each new entry requires manual curation and implementation in order to verify reported dynamics. Our focus on published systems may bias the dataset towards more unusual (and thus reportable) dynamics, particularly because there are infinite possible chaotic systems. Additionally, our model scoring metrics primarily quantify point-wise forecast accuracy; however, over long timescales it may be informative to consider alternative metrics such as stationarity with respect to dynamical properties, or the accuracy of the topology of the predicted attractor [12, 100, 101]. More broadly, we note that in few dimensions chaotic dynamics are rare relative to the space of all possible models [5], although chaos becomes ubiquitous as the number of coupled variables increases [102]. Nonetheless, low-dimensional chaos may represent an instructive step towards understanding complex dynamics in high-dimensional systems.

## Acknowledgments and Disclosure of Funding

We thank Gautam Reddy, Samantha Petti, Brian Matejek, and Yasa Baig for helpful discussions and comments on the manuscript. W. G. was supported by the NSF-Simons Center for Mathematical and Statistical Analysis of Biology at Harvard University, NSF Grant DMS 1764269, and the Harvard FAS Quantitative Biology Initiative. The author declares no competing interests.

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
