# OpenReview forum: "Chaos as an interpretable benchmark for forecasting and data-driven modelling"
_NeurIPS.cc/2021/Track/Datasets_and_Benchmarks/Round2 — NeurIPS 2021 Datasets and Benchmarks Track (Round 2)_

### Official Review · Reviewer_jGaM · 2021-09-19
**Review of the Chaotic Systems Benchmark**

**Rating:** 6
**Confidence:** 3

**Strengths:**

The proposed benchmark is a nice idea of consolidating advances in temporal models on chaotic systems. Typically, most works consider Lorenz or Chua systems and the presence of this benchmark would systematically scale testing of models to a wide variety of systems. Not only is it a consolidated system of many chaotic systems but it also allows for different variations regarding each, eg. presence of brownian noise and granularity of sampling. This could lead to a systematic study of generalization and transfer learning across various variations and prove to be a pretty good test-bed for testing upcoming ML models.

**Weaknesses:**

While the work introduces an interesting dataset and a good set of tasks, I believe that in its current form it is not thorough enough for me to give it a high score. The benchmark provides 131 different dynamical systems along with variations but there is no analysis done on the performance of models across some fundamental different properties. For example, it would be nice to see how performance of different benchmarked models vary on increasing granularity of sampling, or on the presence of brownian noise, or across different fractal dimensions. Such an analysis would definitely make the work stronger and also provide more detailed avenues of where which type of models outperform the others.

**Additional Feedback:**

The authors propose a great benchmark that would help the ML community as a whole. With more experimentation and clear writing, I think it will be a good benchmark for people to start using. I'd be happy to revise my score if some of my concerns are addressed.

**Update**: Increased the score based on the rebuttal provided.

**Clarity:**

I found the paper to be on the harder side to read through and understand well. I think some sections outlining the basics of chaotic systems as well as a dedicated section for interpreting all the results of multiple tasks would make it easier for people to read and understand. It would also be better if the authors use more subsections within the sections to better outline different details of the works (example - Section 2 can be broken into submodules defining variations, mathematical properties, etc. I think an improved structure and more refined writing would make it a much stronger candidate.

**Correctness:**

The dataset is constructed in a sound way and the experimental design is appropriate. The authors consider hyperparameter tuning for each model and dataset and provide ample information regarding the different chaotic systems they use in their work.

**Documentation:**

The authors provide good documentation for their work on their GitHub page.

**Ethics:**

I am not aware of any ethical concerns regarding the dataset.

**Relation To Prior Work:**

It is clearly discussed that the proposed work systematically consolidates different chaotic systems across various fields, thus allowing for better testing of models than considering just one or two chaotic systems like Lorenz.

**Summary And Contributions:**

This work aims to benchmark temporal models of varying complexities for a multitude of chaotic systems across different domains. The paper provides mathematical details regarding each of the chaotic system they use in the benchmark and allows for a number of variations per chaotic system, namely, (a) sampling granularity, (b) initial conditions, (c) multivariate vs univariate state space, and (d) presence or absence of brownian noise in the dynamics. Using this data, the paper constructs a number of tasks like future predictability, model acceleration with importance sampling, transfer learning and symbolic regression.

---

> ### Author Response · Authors · 2021-09-27
> **Response to jGaM**
>
> Thank you very much for your review and feedback! I very much agree regarding the need to illustrate the applicability and failure modes of the different forecasting models, and the need to provide writing that is clear to both the computer science and dynamical systems communities.
>
> I have added all requested experimental results to the revised text, and I have revised the writing. I hope that you will find these changes responsive; please let me know if you have any additional questions or comments.
>
> **[NOISE]** Thank you for this suggestion—I’ve added two new experiments to the SI: a reproduction of all experiments in Fig 2A in the presence of added noise, and a “sweep” experiment inspecting how the ranking of different methods changes as a function of the noise level. I was surprised to find that NBEATS performs very well across a range of noise levels, while the FFT method emerges as the best “simple” baseline in the presence of noise. I have added these results to the supplement, and the main text has been updated in light of these additional findings.
>
> **[GRANULARITY]** I have added a new experiment in the Appendix in which I have retrained all forecasting models across a range of granularities, in order to see how the relative performance of the different forecasting models varies with granularity. Note that both the original forecasting experiment, and the new noise study described above, also include coarse and fine granularity results (shown as the the two sides of each violin in the plots). Interestingly, the relative ranking of different models remains largely the same across granularities, although the performance of the deep learning models fluctuates widely. I speculate that subtle details of the architecture of the DNN models might lead them to having implicit timescale bias, which would likely require extensive network architecture tuning to avoid. I have updated the main text and supplementary discussions in light of these new results.
>
> **[FRACTAL DIMENSION]** I was uncertain how to interpret this request. Since the fractal dimension is intrinsic to each system, I cannot sweep it as I did in the noise or granularity experiments above. However, the current text does include results correlating fractal dimension to forecasting results.
> Please let me know if I am misunderstanding anything, and I will be happy to put together new results.
>
> **[PRESENTATION AND CLARITY]**
> + Per your comments and Ref. pm6d’s suggestion, I have added a glossary to the appendix defining all of the dynamical systems terms used throughout the main text. This portion of the appendix also provides references to several external introductory discussions of the field and terminology.
> + I have restructured Section 2 and demarcated explicit subsections, and I have added an additional subsection that summarizes the high-level takeaways.
> + I added a new section to the discussion of the forecasting task, which focuses on the relevance of these findings to time series practitioners in general. I have also modified the descriptions of the other tasks to provide slightly more general interpretations of the experimental results.

---

> > ### Comment · Reviewer_jGaM · 2021-09-28
> > **Review Update**
> >
> > Thank you for addressing almost all of my concerns as well as adding a number of additional experiments that provide a more holistic view of the benchmark. I also appreciate the better structuring of Section 2 as well as the glossary of terms relevant to the study of chaotic systems.
> >
> > **[Fractal Dimension]**: What I meant was some sort of analysis that shows performance of different algorithms on the Y-axis and the fractal dimensions on the X-axis to see if the same models perform well across all dimensions or the rankings change over fractal dimensions.
> >
> > Nonetheless, given the very useful additions made by the author, I have decided to update my score.

---

> > > ### Author Response · Authors · 2021-09-29
> > > **Made a minor revision to address fractal dimension**
> > >
> > > Thank you very much for reviewing the revision, and for updating your score!
> > >
> > > My apologies that I misunderstood your point about the fractal dimension—I’ve added a new section to the SI where I striate the forecasting results by fractal dimension, entropy, and Lyapunov exponent, to see how different forecasting methods perform when it comes to forecasting the most chaotic vs least chaotic systems.
> > >
> > > Instead of plotting scores vs fractal dimension directly, I decided to group the horizontal axis by quantile in fractal dimension (see new figure in SI). The plots with the points versus score were too difficult to read, since the fractal dimension etc, has a non-uniform distribution across the systems. However, the raw score vs property plots are visible in the forecasting notebook on GitHub.
> > >
> > > I also updated Fig 2 in the main text, to include Spearman correlations between forecasting models and mathematical properties. NBEATS and FFT are the only two models that show meaningful correlation, which is interesting because they were the most consistent models in the noise experiments.

---

> > > > ### Comment · Reviewer_jGaM · 2021-10-03
> > > > **Review Update**
> > > >
> > > > Thank you for the update. This essentially resolves all the concerns I had regarding the benchmark, in light of which I am arguing for acceptance of the work.

---

### Official Review · Reviewer_pm6d · 2021-09-19
**Interesting Dataset with Dual Applications to Chaotic Dynamical Systems and General Time Series Analysis**

**Rating:** 7
**Confidence:** 3

**Strengths:**

&nbsp;

1. The dataset has demonstrable utility for transfer learning on general time series classification tasks, a subject of broad interest to the wider ML community.

2. The author does a good job in highlighting some of the unique properties of chaotic dynamical systems that may encourage the development of bespoke methods for forecasting in this domain, an example of which being the application of importance sampling.

3. The dataset is clearly documented with a full list of all systems comprising the dataset described in the supplementary material. The codebase is in general well documented (see minor comments below).

4. Ideas for future applications are dispersed throughout the paper.

5. The paper is well written and is, in general, understandable by a lay machine learning audience (see minor comments below).

&nbsp;

**Weaknesses:**

&nbsp;

## __MAJOR POINTS__

&nbsp;

1. It may be helpful to provide a **glossary of terms** (perhaps in the SI) including basic definitions of terms such as attractor (e.g. set of states to which a system evolves), strange attractor (attractor exhibiting fractal structure) etc. This should help make the subject matter more accessible to a lay machine learning audience.

2. Some explanation of the links between strange attractors and chaotic systems may be beneficial in the introduction e.g. the existence of **strange attractors which are not chaotic** [1].

3. In relation to Task 1: Forecasting, while the findings are interesting, it may be helpful to users to provide some ideas about how to use this task as a benchmark for developing models that are appropriate for forecasting in chaotic dynamical systems?

4. For Task 4: Data-driven model inference and prediction, is the implication that the benchmark should be extended to be more challenging through the inclusion of aspects such as measurement noise? The implication seems to be that the chaotic systems benchmark in its current form is too simple to differentiate between state-of-the-art models?

&nbsp;

## __MINOR POINTS__

&nbsp;

1) Renaming the sections may make the paper easier to read e.g. having section 3.1 as the conclusion in a separate section 4.

2) Increasing the fontsize of figure 4 for example, should be possible given the extra page for the camera-ready version.

3) How are the errobands in Figure 3 computed?

4) Table 1, how are the standard errors computed?

5) Task 3 appears to refer to a Table 3 that doesn't exist?

6) The readability of Table 1 would be improved if the task was defined in the caption alongside a description of how the standard errors were computed.

7) In the conclusion, it is stated that importance sampling is an application of the dataset but is this limited to applications involving chaotic attractors? The use of transfer learning for general time series classification appears to be more general.

8) In the section on related work, is it necessary to cite general time series benchmarks? It seems that this literature may be too large to provide a detailed account of all benchmarks e.g. MIMIC [2] in healthcare for example.

&nbsp;

## __REFERENCES__

&nbsp;

1) Grebogi et al., Strange attractors that are not chaotic. Physica D: Nonlinear Phenomena. 1984.

2) Johnson et al., MIMIC-III, a freely accessible critical care database. Scientific Data. 2016.

&nbsp;

**Additional Feedback:**

&nbsp;

All points covered in the main response.

&nbsp;

**Clarity:**

&nbsp;

Overall very good, minor points noted above.

&nbsp;

**Correctness:**

&nbsp;

To the best of my knowledge the results of the author appear to be correct. The statistical hypothesis tests (under the assumption of appropriate hyper-parameter tuning) lend credence to the claims made by the author.

&nbsp;

**Documentation:**

&nbsp;

The demo notebook is useful, a datasheet specification is provided in the SI. The usage section of the README is informative. There appears to be scope to improve the presentation of some of the supplied notebooks e.g. comments could be added with removal of sections of the code that appear to be used for debugging purposes.

&nbsp;

**Ethics:**

&nbsp;

Adequately addressed in the SI.

&nbsp;

**Relation To Prior Work:**

&nbsp;

Adequately addressed.

&nbsp;

**Summary And Contributions:**

&nbsp;

The paper introduces a set of 131 chaotic dynamical systems spanning application domains including climatology, astrophysics and biochemistry. The author makes a strong case for the niche which the dataset fills within the canon of literature on general time series analysis; through illustrative experiments the author showcases some unique properties of chaotic dynamical systems whilst also highlighting use-cases for the dataset as a general tool for transfer learning and data augmentation.

I believe the dataset is an interesting and useful contribution to the machine learning community and as such veer on the side of acceptance. I have a few minor concerns which, if addressed in the rebuttal, may warrant an increased score.

&nbsp;

---

> ### Author Response · Authors · 2021-09-27
> **Response to pm6d**
>
> Thank you very much for your review and recommendations on improving the manuscript! In the revision, I have sought to address all of your points. Please let me know if you have any additional questions or comments.
>
> **[MAJOR POINT 1 - Glossary and accessibility]** Thank you very much for this recommendation, I have added a glossary to the appendix, and I have added a brief summary of a few key terms in the main text. I have also updated the structure of the Description section to make discussion clearer to those without a dynamics background.
>
> **[MAJOR POINT 2 - Non-chaotic systems]** I agree that this needs to be more explicit, and in the revised Description section I clarify that that included systems are both chaotic and fractal as independent properties. The revised section now references non-chaotic strange attractors, as well as systems that are chaotic but not attractors (e.g. transient chaos or chaotic repellers)
>
> **[MAJOR POINT 3 - Model development]** I have added a conclusion to the forecasting section that summarizes the high-level takeaways for forecasting practitioners. Briefly, I note that hierarchical models (like NBEATS or TCN) are likely a good choice for multiscale systems (or systems with broadly continuous spectra, like chaotic systems). I also note that these results agree with recent works on Koopman operators that suggest that lifting a nonlinear system to a high dimensional space (e.g. via kernels) can make the dynamics more linear and thus predictable—and that these methods likely outperform linear liftings like time delay embeddings.
>
> **[MAJOR POINT 4 - Regression task difficulty]**  I agree that the problems appear to be relatively easy, and that the hardest problems are those contain expressions and functions unlikely to appear in a standard function library. However, I do not think that this issue is unique to this chaotic systems database: the common Nguyen dataset (which only contains 12 expressions) and the Feynman dataset are both relatively simple sets of problems. In particular, the Feynman equations are dimensional, which makes regression even easier (an interesting discussion of this can be found in the reviews for the DSR paper at ICLR last year). The newly-released SRBench does contain harder problems in their test suite, but my understanding is that their database includes general time series problems for which closed-form analytical formulae may not exist.
>
> I would prefer to defer noisy symbolic regression to future studies, since there are some subtleties I would need to address in order to properly benchmark it: not all symbolic regression methods include regularization off-the-shelf, and non-deterministic components could affect the dynamics in unexpected ways, such as via noise-induced chaos or stochastic resonance. A more straightforward way to make harder problems would potentially be to nest or couple existing chaotic systems together, although that is beyond the scope of the current work.
>
>
> **[MINOR POINTS]**
>
> Minor Points 1-7 have all been fixed, thank you very much for identifying these issues.
>
> Minor Point 8: Regarding the related work section, I agree that it is not feasible for the paper to review all previous time series benchmarks. My general heuristic for this section was to mention the main datasets used in the papers describing the various forecasting methods, but I recognize that that may not be representative.
> For now, I would prefer to keep this paragraph, since it establishes the typical scale of existing time series collections; however if the referee feels that this detracts from the writing, I would be happy to remove it or put it in the SI.
>
> Documentation: I agree that the debugging code in the benchmark notebooks is distracting, and I have removed commented-out cells. I did keep a few minor comments for making plots, etc, but the large blocks of commented-out code have been removed.

---

> > ### Comment · Reviewer_pm6d · 2021-09-30
> > **Many Thanks for the Paper Revisions and Codebase Modifications; Upgraded Score**
> >
> > &nbsp;
> >
> > Many thanks for including the glossary of terms as well as the clarification on non-chaotic systems! The point on regression task difficulty was not intended as a criticism and indeed the dataset should serve as an extensible platform for other interested researchers to construct more complex tasks! Ideas for how this might be performed are now clarified and any further suggestions from the author would be welcome in the manuscript.
> >
> > &nbsp;
> >
> > Following the author response, I upgrade my score.
> >
> > &nbsp;

---

### Official Review · Reviewer_SPdr · 2021-09-22
**Cha**

**Rating:** 8
**Confidence:** 3
**Correctness:** Yes
**Clarity:** Yes

**Strengths:**

- The paper creates a huge dataset repository that can easily be extended.
- The paper compares multiple methods for time series forecasting of chaotic systems and show that Neural Networks tend to outperform statical and traditional methods on this particular task.
- The paper shows that the datasets can be used for transfer learning on different classification tasks for UCR dataset (this is my favorite contribution) since the usage of chaotic time series to improve the accuracy of non chaotic is novel and has many useful applications.
- The paper shows how training can be accelerated by upsampling sparser regions which reduced the time and  error compared to training full epoch.
- The paper showed that this dataset can be utilized for data-driven modeling in the form of a symbolic regression task.

**Weaknesses:**

- The usage of term interpretable might not be misleading, the data itself is interpretable due to its mathematical properties however it us unclear how this data can improve or benchmark model interpretability in time series [1].
-  As mentioned in the paper, all included chaotic systems that have previously appeared in published works.



[1]Ismail, Aya Abdelsalam, et al. "Benchmarking deep learning interpretability in time series predictions." arXiv preprint arXiv:2010.13924 (2020).

**Additional Feedback:**

Minnor comments:
Line 292: I think you meant table 1 instead of table 3

**Documentation:**

The code is well documented and reproducible

**Relation To Prior Work:**

Yes

**Summary And Contributions:**

The paper introduces a dataset consisting  of 131 known chaotic dynamical systems. The paper benchmarks the performance of 16 forecasting models including deep learning, statical, classical and machine learning methods on forecasting chaotic time series task. The paper shows the usefulness of dataset in transfer learning of to improve the classification accuracy of UCR datasets. The paper shows that training can be accelerated by encouraging the network to oversample sparser regions of the underlying attractor.  Finally the paper use the proposed dataset for data-driven modeling in the form of a symbolic regression task.

---

> ### Author Response · Authors · 2021-09-27
> **Response to SPdr**
>
> Thank you very much for your positive review and suggestions! I have modified the first section of the main text to clarify interpretability in the context of this paper, and included references to prior work on saliency and feature selection for time series. I have also modified sections of the dataset description in order to improve the clarity of some the dataset and results are presented. Please let me know if you have any additional questions or comments.

---

### Official Review · Reviewer_b7Q2 · 2021-09-22
**Dynamics systems benchmarks**

**Rating:** 8
**Confidence:** 4
**Correctness:** The dataset is soundly constructed.

**Strengths:**

This dataset seems great for benchmarking Ml models and deep learning models which can model dynamics continuously or discretely. As their tests show this can include recurrent neural networks, transformers and other statistical model.  Because of known ground truth one can test interpretability and ability to reverse engineer or identify systems as well.

**Weaknesses:**

The suite uses chaotic dynamic systems which may be hard for ML methods in general, non-chaotic dynamics could be included as well in future versions.

I would have liked to see a test on neural ODE systems, these seem ideally suited for this benchmark dataset.

The github has a bunch of todos particularly with regards to lyapunov exponent calculation, hopefully these can be addressed soon.

**Additional Feedback:**

I would include some non-chaotic systems as well.

**Clarity:**

The paper is well written. One could clarify how these test cases could aid the development of machine learning models. Presumably chaos prediction does not always have to be the end goal, but rather this could be a test of dynamic neural network capabilities.

**Documentation:**

The code repository is easy to understand. The visualizations are helpful.

**Ethics:**

None.

**Relation To Prior Work:**

Previous work is adequately discussed.

**Summary And Contributions:**

This work contributes 131 different chaotic dynamic system simulations and 16 precomputed simulations. It also includes code for resimulation and computation of features of the dynamic system such as lyapunov exponent.

---

> ### Author Response · Authors · 2021-09-27
> **Response to b7Q2**
>
> Thank you very much for your positive review and comments! I very much appreciate your advice and interest in the paper. I have made several changes (listed below) that may be of interest; please let me know if you have any additional questions or feedback.
>
> **[NEURAL ODE]**  I agree that these are a great fit for this dataset. I have added a preliminary neural ODE experiment to the supplementary information and benchmarks repository, and I have briefly discussed the experiment in the main text. Interestingly, I did not observe a correlation between dynamical system properties and nODE model quality, but I believe that deeper investigation would require a separate, dedicated study. In any case, I hope that my nODE example code (which is now in the GitHub repository) will be a useful starting point for practitioners.
>
> **[MODEL DEVELOPMENT]** I have updated the forecasting section with a new conclusion, describing the high-level takeaways for forecasting practitioners, based on the chaotic systems results.
>
> **[GITHUB FUTURE]**  I agree, and several of these goals (e.g. delay equation calculations) have been added. The development bottleneck right now is creating a separate database of Jacobian matrices for each dynamical system—these can be used to improve Lyapunov exponent calculations, as well as accelerate numerical integration. However, these need to be computed analytically by hand, which has been challenging.
> Right now, I’m prioritizing extending the “maps” module, since discrete-time maps have unique properties (discontinuous dynamics, reversibility) that differentiate them from continuous-time systems. I’ll prioritize other extensions based on usage patterns of the dataset.

---

### Author Response · Authors · 2021-09-27
**Revision Uploaded -- Thank you!**

Thank you very much to the four reviewers for their comments and suggestions! I have posted a revised manuscript addressing all comments and requests for additional experiments. I think that the manuscript has really improved as a result.

I have responded to each referee individually below; here I summarize the major changes:

**[EXPERIMENTS]**
+ Replicate of the full forecasting task with noise, showing that the general results are robust (jGaM)
+ Relative performance of different forecasting models as noise and granularity are varied (jGaM)
+ A neural ODE task in the SI, showing baseline performance of nODE across the dataset (b7Q2)

**[EXPOSITION]**
+ New glossary defining key terms and concepts from dynamical systems theory. A summary of major terms, and a reference to the appendix, now appears in the main text (pm6d)
+ I restructured and re-wrote portions of the “Description” section to have clearer demarcations between sub-topics. I also re-worded portions of the section introduction in order to make the results a bit clearer. (jGaM)
+ I added a subsection to the forecasting results, describing potential model-building implications for forecasting practitioners (b7Q2 & pm6d)

Please let me know if you have any additional comments or requests.

---

> ### Author Response · Authors · 2021-09-29
> **Minor Update**
>
> Based on further discussion with Referee jGaM, I have made additions to Figure 2 and added a section to the SI. The major changes described above remain the same. Thanks again for your comments and feedback!

---

### Comment · Area_Chair_kcDN · 2021-09-28
**Discussion/Acknowledgement**

Thank you all for your reviews, authors' response, and the revision! For those reviewers who have asked concrete questions - please respond to the thread with acknowledging authors' response, and raise any further questions that wasn't resolved by the response.

---

### Decision · Program_Chairs · 2021-10-09

**Decision:**

Accept

**Comment:**

All reviewers agreed that this dataset makes good contribution. It includes many known chaotic dynamical systems, benchmarking 16 forecasting models with many ML methods (not just DL). It show cases usefulness in transfer learning, symbolic regression and others. Authors has responded to reviewers’ comments appropriately (eg adding glossary terms).